# From Ground to Glass: Evaluation of Unique Barley Varieties for Craft Malting, Craft Brewing, and Consumer Sensory

**Evan B. Craine [1], Stephen Bramwell [2], Carolyn F. Ross [3] and Kevin M. Murphy [1,*]**

[1] Department of Crop and Soil Science, Washington State University, Pullman, WA 99163, USA; evan.craine@wsu.edu

[2] Thurston County Extension, Washington State University, Lacey, WA 98506, USA; stephen.bramwell@wsu.edu

[3] School of Food Science, Washington State University, Pullman, WA 99163, USA; cfross@wsu.edu

[*] Correspondence: kmurphy2@wsu.edu

**Abstract:** Differentiating agricultural products has been adopted as a strategy to improve farm profitability and thereby business sustainability. This study aimed to evaluate unique barley varieties for craft malting and brewing markets to enhance profitability for diversified grain growers in southwestern Washington. Advanced barley breeding lines from Washington State University (WSU) were compared to a control variety (CDC-Copeland) through field trials, experimental and commercial malting conditions, and consumer sensory evaluation. The beers differed only by the genotype-dependent malt. Malting conditions (experimental or commercial) and field replicate influenced five out of the eight malt quality traits measured, while genotype influenced seven out of eight of the traits. Consumers differentiated the beers through ranking, open description, and check all that apply during a central location test. Based on consumer liking, breeding lines 12WA_120.14 or 12WA_120.17 could replace CDC-Copeland in beers. A total of 83% of consumers responded that they would pay more for a beer if it would support local farmers. This value proposition represents an opportunity for consumer purchasing to support producers, who form the foundation of the craft malt and beer value chain and whose economic success will determine the sustainability of small farms in minor growing regions. We provide further evidence for the contributions of barley genotype to beer flavor, while tracing the impact of barley genotype from ground to glass.

**Keywords:** value added; barley; malt; beer; consumer sensory; regional grain systems; craft; flavor; micromalting

## 1. Introduction

Differentiating agricultural products, whether based on production system, alternative supply chains, flavor, or other value propositions, has been adopted as a strategy to improve farm profitability. Applications of this approach are organic certification, Community Supported Agriculture (CSA) subscription box programs [1], local meat marketing [2], values-based supply chains [3], and distinctive flavors in craft food and beverages, among other approaches [4,5]. The emergence of these strategies has required the development of objective and subjective means of validating product differences. These paired differentiation-validation pathways represent sources of opportunity and creativity for those working to rebuild value around agricultural producers, in a global food system context that has drained value away from the production centers of raw agricultural products. As efforts to reconstruct value around agricultural producers continue to evolve and mature, opportunities to build frameworks for product differentiation in local food systems will expand, along with their complimentary validation methods. A renewed interest in the development and strengthening of regional grain systems is leading to a relocalization of staple crops in non-traditional growing areas [6–8].

### 1.1. Increased Attention to Malt Differentiation

Specific frameworks exist within public plant breeding and product development as a fundamental effort to differentiate, or distinguish, farm *goods* from conventional commodity *products*, by which the true nature and costs of production are transparent and even highlighted in end products [9]. Product differentiation, as well as value-based supply chains, are important strategies that can enhance the profitability and sustainability of agricultural enterprises [10]. As grower and consumer experience with varied labeling and other value schemes increases, so too have opportunities to restructure cereal grain product patterns in highly consolidated industries, such as malting and brewing, and increase awareness of sustainability related to grain production [11]. Consolidation of breweries and malthouses, driven by efficiencies in production and distribution, led to just 4 and 10 companies producing over one-half of the global supply of beer and malt, respectively, by 2012 [12–14]. This consolidation restricted styles and flavors to a narrow range of mass-produced adjunct beers, which may have led to homogenization of consumer preferences [15], while at the same time constricting malt contracts to fewer buyers [16]. As a result, barley breeders have been constrained during germplasm development to stringent quality parameters. Combined with a focus on yield and disease resistance, barley breeding programs have relied on crossing elite by elite material to improve varieties, which has effectively narrowed the genetic base of varieties and reduced the capacity to adapt to new abiotic and biotic stressors that threaten yield and quality [17,18]. The result regarding the economics of farm goods is a valuation of malting barley at 0.05 USD, in a typical 6-pack of beer costing 8.99 USD on average [19].

However, the rise of all-malt, craft brewing, and the current rise of craft malting, has supported a shift in consumer preferences to novel styles and flavors, a paradigm shift in defining malt quality, a novel economic opportunity for barley growers and a novel breeding opportunity for barley breeders. If, and when, U.S. craft brewers attain 20% of market share by volume, they are projected to use over 50% of the malt used by U.S. brewers [20,21]. Market share increased from 13.0% in 2018 to 13.6% in 2019, but fell to 12.3% in 2020 amid the coronavirus pandemic [22]. Craft malting, brewing and distilling industries, endeavoring as they are to connect breeders, farmers, end-users, and consumers, have an opportunity to increase both diversity of products (in flavor attributes and germplasm) and mechanisms for price recovery for producers through integrated approaches to sustainable development. In 2020, Craft Maltsters Guild members reported paying barley producers 0.37 USD/kg for malting barley, compared to 0.18 USD/kg as reported by the Idaho Barley Commission for Idaho Falls [23,24].

Consumers are increasingly concerned with where a product comes from and how it is produced. Moreover, consumers may find virtue in supporting those that produce and craft products and are willing to pay a premium to reward their skill and hard work [4]. As with bourbon, chocolate, and cheese, "third-wave" coffee is defined by de-commodification and an increase in singularity phrasing and has emerged in the context of high-end artisan food and beverage markets [5]. Changes in the consumer perception of craft malt and beer can be better understood in comparison to these industries, which have been transformed in part by this the process of decommodification. While the current "wave" of the craft brewing and distilling industries has not been clearly defined, they are composed of various associations that strive to connect producers of raw ingredients with maltsters, brewers and distillers, and consumers. In some cases, identification of product origin to specific farms is carried all the way through to marketing, branding, and labeling. These efforts demonstrate the ability to alter production patterns, while enhancing consumer education and increasing awareness of sustainability related to grain production.

### 1.2. Barley Genotype as a Driver of Malt Differentiation

In 2013, the Craft Maltsters Guild was created to support the craft malt "revolution". Craft malt is objectively differentiated from commodity malt as defined by the Guild: small (5 metric tons to 10,000 metric tons of malthouse production per year), local (50% of grains

sourced within 500 miles of the malthouse) and independent (malthouse independently owned by 76% majority). The Guild organizes a yearly conference with academic, business, and trade presentations. In 2017, Guild members established a standardized method, approved by the American Society of Brewing Chemists (ASBC), for the evaluation of extractable malt flavor, to support objective validation of craft malt differentiation [25]. Moreover, the sensory software company DraughtLab and ASBC collaborated to develop base malt and specialty malt flavor maps, which provide a lexicon and visual aid for sensory evaluation.

Several studies have recently validated the important contributions of barley genotype, such as experimental breeding lines and commercially released varieties, to malt and beer flavor differentiation. Both genotype and the environment in which the barley is produced can influence beer flavor [26], and genotype can have the strongest influence regardless of how well modified the malt is [27]. Genotype can also influence consumer acceptance, or beer liking, and specific sensory attributes can drive consumer preferences [28–30]. These sensory attributes are in turn partially related to specific chemical compounds within beer metabolomic profiles [28,29,31]. This emerging body of evidence provides a framework for barley breeders who aim to maintain or improve malting quality for brewers, while working with brewers to redefine malt quality to include flavor discovery, and to meet consumer demands with strategic variety development.

Guided by insights into consumer preferences, barley breeders can more effectively target beer flavor. The millennial generation, consisting of approximately 70 million consumers from 28 to 38 years old and making up the largest generation in the U.S, is a major driver of shifting consumer beer preferences stemming from the rise of craft beer and the "taste revolution" [32]. Within craft beer, millennials account for 85% of consumption capitol. As the craft beer market continues to differentiate, 30% of the market is predicted to shift to young consumers, whose preferences will be shaped by the plethora of options available as they turn 21 years old [33]. These consumers are characterized by being adventurous in their decisions, prioritizing flavor and freshness over price, and valuing sustainability and "local" [34]. In general, craft beer consumers value sustainability, with the majority willing to pay more for sustainably produced beer [35]. Moreover, craft beer consumers prefer beer produced by locally owned and independent breweries and are willing to pay more for these attributes [36].

To meet rising demand and decentralize the malting and brewing supply chain, regional grain systems are evolving via enhanced farmer–maltster–brewer–consumer relationships. Malt barley breeders have the opportunity to support this effort, as they identify and balance the needs of farmers, maltsters, brewers, and consumers to develop regionally adapted malting barley varieties, with excellent malt quality and unique and distinctive flavors [9]. This study builds upon the work of Craine et al., 2021 [30], and together these studies are part of a larger project with the long-term aim of creating high-value grain markets in southwestern Washington to maintain profitability of agricultural enterprises imperiled by rapid population growth and unsustainable patterns of development. Craine et al., 2021 [30] found that untrained, craft beer consumers had the ability to differentiate beers made from experimental barley breeding lines and a control variety, malted to similar specifications, using a 9-point hedonic scale and the check-all-that-apply (CATA) methodology.

To advance the research, this study aimed to explore the ability of untrained, craft beer consumers to differentiate beers made using the same breeding lines, except for one released as the variety Palmer (formerly 11WA-107.43), and the same control variety, by ranking the beers based on overall liking, providing a description in their own words, and selecting attributes from a beer flavor map during a central location test. Consumers also completed a survey to gauge their involvement with the regional grain system and willingness to support barley producers through purchasing and consumption of craft beer. In addition to the consumer sensory evaluation, this study also aimed to evaluate the barley varieties in the context of small-scale, experimental malting as well as commercial-scale malting,

and a replicated field trial conducted in southwestern Washington. Evaluating the barley varieties from ground to glass provided a systems-based approach to understanding how various factors along the supply chain interact to influence grain, malt, and beer quality, and provided a more comprehensive understanding of the challenges associated with the development of regional grain systems.

## 2. Materials and Methods

### 2.1. Germplasm and Grain Production

Three spring, two-row genotypes from the WSU barley breeding program and a control variety (CDC Copeland, Crop Development Center, University of Saskatoon; hereafter Copeland) were grown in Adna, WA in 2018. Breeding lines included 10WA_117.17 (Radiant/Baronesse/3/WA 10701-99//Baronesse/Harrington (X06G10)/4/Pmut-422H/CDC Candle (05WA-344.1); hereafter 117.17), 12WA_120.14 and 12WA_120.17 (WA 10701-99/NZDK 00-131//AC Metcalfe; hereafter 120.14 and 120.17, respectively). Barley was planted in a randomized complete-block design with three replicates on 27 April 2018 and harvested on 10 and 13 August 2018.

### 2.2. Grain and Malt Quality

Biological replicates (*n* = 3) of each genotype from the field trial were malted on an experimental scale using a uniform method (performed by the United States Department of Agriculture [USDA] Cereal Crops Research Unit [CCRU], Madison, WI, USA) and a modified method (performed by Hartwick College Center for Craft Food and Beverage, Oneonta, NY, USA) to develop recommendations for ideal malting conditions at scale. The uniform method used a standardized malting regime and the modified method used genotype-dependent steep conditions (Table S1). Approximately 1800 kg of each genotype was malted by the craft malthouse Gold Rush LLC (Baker City, OR, USA) (Table S2). The resulting malt was analyzed by the Hartwick College Center for Craft Food and Beverage (Oneonta, NY, USA). All malt quality analyses were performed using official methods of the American Society of Brewing Chemists (ASBC). Malt quality traits include friability (ASBC Malt-12), extract (ASBC Malt-4), color (ASBC Wort-9), β-Glucans content (ASBC Wort-18B), soluble to total protein ratio (S/T; the Kolbach index) (ASBC Wort-17), free amino nitrogen content (FAN) (ASBC Wort-12), diastatic power (ASBC Malt-6C), and alpha-amylase (ASBC Malt-7C).

### 2.3. Beer Sample Preparation

The beers were developed by Top Rung Brewing Company (Lacey, WA, USA) to be "malt-forward" to showcase the malt, and the beers differed only by the genotype-dependent malt used. Each beer was stored in a keg on ice, and samples were gently poured immediately before serving during the consumer sensory evaluation.

### 2.4. Consumer Sensory Evaluation

This study was reviewed and approved by the Washington State University Institutional Review Boards (IRB #17775) and informed consent was obtained from subjects prior to their participation in this study. The consumer sensory evaluation was conducted as a central location test [37]. The panel (*n* = 138) was comprised of untrained panelists (hereafter consumers) recruited during the Tumwater Artesian Brewfest, an agritourism event, on 17 August 2019 (Tumwater, WA, USA). The consumers attending the event were generally craft beer enthusiasts, and while at the event, heard of the sensory evaluation by word of mouth, by viewing posted signs, or by viewing consumers engaged in the sensory evaluation. All consumers were screened prior to participating. Individuals were disqualified from participating if they were not over 21 years of age, had a gluten intolerance, were pregnant or expected that they were pregnant, or were visibly intoxicated. Participants were required to complete a survey prior to the evaluation, which consisted of 10 questions related to demographics, consumption patterns questions, factors that

influence beer purchasing, and favorite beer styles and brands. The questions and possible responses for the survey are provided in Table S3. Consumers received a non-monetary compensation for their participation.

The evaluation was conducted outside in a designated area of the event space, at a maximum distance from other food and beverage vendors to limit distractions and confounding aromas. Eight sensory booths were constructed using a series of 2.4 m tables with dividers. The four different beer samples (30 mL; one for each genotype) were each assigned a 3-digit blinding code and presented to each consumer simultaneously. Consumers were instructed to evaluate samples in the assigned order; serving orders were randomized and balanced. Samples were kept on ice and poured immediately before serving to maintain temperatures close to 4 °C. Each sample was presented in a clear, 118 mL beer tasting glass. A cup of distilled water and two saltine crackers were provided as palate cleansers, along with a cuspidor. Consumers were instructed to taste one sample at a time and cleanse their palate between samples.

For each sample, consumers provided a short description in their own words (i.e., open-ended description), ranked the samples in their order of preference based on overall liking (1 = like the most, 4 = like the least) considering the sample appearance, flavor/aroma, mouthfeel, and aftertaste, and were asked to choose as many attributes as they would like from the provided beer flavor map (https://www.draughtlab.com/flavormaps; accessed on 17 July 2019) that would best differentiate the sample (i.e., check all that apply; CATA). Responses were grouped and analyzed according to the following data sets: "Open Description" (open-ended description coded according to the terms represented on the beer flavor map), "Novel Terms" (novel terms from the open-ended description not included in the beer flavor map), "CATA" (check all that apply using the beer flavor map), and "Combined" (CATA and Open Description data sets combined). These categories were formed to provide structure to the data set originating from the consumers responses, and to provide insights into similarities and difference between response types. The open-ended description was intended to provide consumers with the freedom to self-select terms from their own vocabulary and communicate their experience based on their respective reference points.

The CATA task using the beer flavor map was intended to guide the consumers, by providing a collection of possible terms, while still providing flexibility with sample description. The beer flavor map is a novel tool developed by DraughtLab to meet a need for a standardized lexicon within beer sensory evaluation. Designed to be accessible by brewing professionals and consumers alike, the flavor map is comprised of sensory terms encompassing taste, aroma, and mouthfeel and is designed as a visual tool and intuitive memory aid to help identify and describe a wide range of possible flavors. Craine et al., 2021 [30] selected CATA attributes from the beer flavor map and consumers demonstrated the ability to use these terms to differentiate genotype-dependent beer samples. The sensory methods used in this study were selected to build on this research by further testing the ability of craft beer consumers to effectively communicate their experiences.

### 2.5. Statistical Analyses

All statistical analyses were conducted in XLSTAT Version 2020.5.1 (Addinsoft, Paris, France), unless otherwise noted. Rank data were analyzed using the non-parametric Friedman's test, with post hoc analysis consisting of multiple pairwise comparisons using Nemenyi's procedure. Correspondence analysis (CA) was performed and biplots were generated using the *FactoMineR* package in the R statistical software (hereafter R) [38]. Attributes that generally had at least five citations per genotype were included in analyses, resulting in a filtered list of attributes from the primary data set generated. Content analysis was used with the open description data to code the attributes according to the beer flavor map, and to identify novel attributes, while the flavor map data were grouped according to the primary attributes or themes represented on the beer flavor map. Cochran's Q test tested for significant differences among genotypes for each attribute, and multiple pairwise

comparisons using the critical difference (Sheskin) procedure allowed for separation of the genotypes by attribute. Malt quality from the replicated field samples was analyzed using a three-way ANOVA, with malt method (uniform or modified), genotype and field replicate as random effects, and mean separation was performed using Fisher's Least Significant Difference (LSD) with Bonferroni-corrected *p*-values using the agricolae package in R [39]. Correlations were performed using rcorr in R [40]. Principal component analysis was performed using the prcomp function in the stats package in R [41]. Variables were shifted to zero centered and scaled to have unit variance. Ordered probit models were generated using polr in the MASS package in R [42].

## 3. Results and Discussion

### 3.1. Factors Influencing Beer Preference and Panel Composition

Based on the pre-tasting survey, we found that the factors that the consumers used to select beers differed significantly (Cochran's Q Test, Q Observed = 173.3, df = 8, *p* < 0.0001). Beer style and unique flavors, cited by 49.3% and 47.8% of consumers, respectively, were the most-often cited determinants of beer selection (Critical difference [Sheskin] procedure). These two selection criteria were used more frequently by greater margins than other criteria, including season or weather (27.5%), location of brewery (22.5%), price (18.8%), notable ingredients (i.e., local, organic, or other) (15.9%), brand (10.1%), or label (8.0%). The identification of beer style and flavor at essentially equal rates is not surprising given the close link between them. A review by Betancur et al., 2020 [43] discusses these selection factors in combination (reflecting a close relationship for example between style and aroma) while also noting links between flavor preferences and factors such as nationality, ethnicity, and gender. As a result, flavor as a selection criterion may relate closely to other factors, such as notable ingredients. Research by Donadini and Porretta (2017) [44] indicate that novel ingredients were associated with "unusual sensory notes", while work by Jaeger et al., 2020 [45] suggests that less notable attributes are currently lost among two prevailing craft beer segments, consisting of those preferring strongly flavored as compared to mildly flavored beers (i.e., IPAs versus lagers). Therefore, notable ingredients may need to be distinguished from their downstream flavor impacts. Craft beer consumer preferences are increasingly influenced by discovering distinctive flavors, and craft beer consumers can be adept at objectively communicating their varying experiences [32,44,46].

Panel composition is an important consideration in flavor evaluation, given the implicit outcome of evaluation as an appraisal of value. Evaluating flavor raises questions such as 'Who is tasting', 'how do we taste differently', 'what is tasted', and 'how are different flavors valued'. Consumers in this study self-selected by attending the Tumwater Artesian Brewfest, and generally represent prevailing demographic patterns among craft beer drinkers, with some exceptions. Compared to the average craft beer drinker population [47], this panel enjoyed greater female representation (52.2% as compared to 31.5%) and average minority representation (14% as compared to 13.4%). Age ranged from 21 to 61+, with the majority in the 21–30 (24%) and 31–40 (26%) age ranges. Panel characteristics are defined in Table S3.

While we were limited in our ability to evaluate the effect of demographics on beer liking and evaluation of flavor attributes, this research (particularly as it involves selection of "preferred" flavor attributes) must be mindful of panel composition and potential effects on sensory evaluation, which have been described by Lahne (2018) [48], Lahne and Spackman (2018) [49], and Berenstein (2018) [50].

### 3.2. Barley Genotype Influences Grain and Malt Quality

While consumer sensory evaluation can be highly variable due to differences in preferences and expectations, as well as perception and reporting of attributes, malt quality analysis affords a more objective evaluation that is perhaps most relevant to brewers. Therefore, we utilized two different malting regimes through small-scale, experimental malting to evaluate samples from the replicated field trials for malt quality, and used this informa-

tion to help inform the commercial maltster's decision making. Within the experimental malting, the modified method adjusted germination conditions to each genotype compared to the uniform malting conditions.

Industry guidelines, published by the American Malting Barley Association (AMBA) and Brewers Association (BA), are intended to ensure sufficient malt quality for the all malt craft brewing industry, while providing targets for barley breeders and maltsters. Malt produced using the uniform method failed to meet any of the BA or AMBA malt quality guidelines, except for diastatic power (DP) (Table 1). For the malts produced using the modified method, only DP and alpha amylase were within the target ranges. Overall, samples produced using the modified method had malt quality much closer to industry guidelines, compared to samples produced using the uniform method.

**Table 1.** Summary statistics, including the mean and the standard error of the mean (SEM) reported for malt quality traits overall, and for each malt method, genotype and replicate, respectively. Malt methods include uniform (all genotypes malted under the same conditions) and modified (genotypes malted according to genotype-dependent malting conditions). Mean separation performed separately for malt method, genotype, and field replicate, using Fisher's Least Significant Difference (LSD) with Bonferroni-corrected *p*-values. For the columns in malt method, genotype and replicate, values within each row that do not share a letter are significantly different.

| Malt Quality Trait | Maltster | | Genotype | | | | Replicate | | | Overall | BA [1] | AMBA [2] |
|---|---|---|---|---|---|---|---|---|---|---|---|---|
| | Uniform | Modifed | 117.17 | 120.14 | 120.17 | Copeland | 1 | 2 | 3 | *n* = 24 | | |
| Barley Protein (%, DB) | 12.2 a ± 0.07 | 12.0 a ± 0.08 | 11.7 b ± 0.15 | 12.7 a ± 0.06 | 12.6 a ± 0.14 | 11.5 b ± 0.12 | 11.5 b ± 0.12 | 12.9 a ± 0.06 | 12.0 b ± 0.05 | 12.1 ± 0.04 | | |
| PLMP (on 6/64") | 93.4 a ± 0.13 | 92.8 a ± 0.24 | 91.2 a ± 0.58 | 94.5 a ± 0.23 | 93.2 a ± 0.28 | 93.5 a ± 0.16 | 92.7 a ± 0.12 | 93.2 a ± 0.26 | 93.4 a ± 0.43 | 93.1 ± 0.1 | | |
| FGDB (%) | 80.1 a ± 0.07 | 80.3 a ± 0.07 | 79.5 b ± 0.11 | 80.2 ab ± 0.1 | 80.2 ab ± 0.11 | 80.9 a ± 0.13 | 80.6 a ± 0.11 | 79.6 b ± 0.08 | 80.3 ab ± 0.08 | 80.2 ± 0.03 | | >81.0 |
| KI (%) | 48.3 a ± 0.39 | 45.5 b ± 0.44 | 40.9 a ± 0.61 | 47.2 b ± 0.37 | 46.5 b ± 0.4 | 53 c ± 0.42 | 48.6 a ± 0.6 | 44.8 b ± 0.62 | 47.3 a ± 0.67 | 46.9 ± 0.21 | 35–45 | 38–45 |
| β-glucan (ppm) | 412 a ± 11.48 | 187 b ± 4.78 | 432 a ± 34 | 313 b ± 20.83 | 213 c ± 14.94 | 240 bc ± 16.94 | 276 a ± 18.88 | 316 a ± 22.89 | 306 a ± 18.27 | 299 ± 6.46 | <140 | <100 |
| DP (°L.) | 98.8 b ± 1.14 | 118 a ± 1.09 | 94.7 b ± 2.09 | 113 a ± 2.28 | 119 a ± 3.06 | 108 a ± 2.23 | 102 b ± 1.93 | 119 a ± 2.03 | 105 b ± 1.74 | 109 ± 0.67 | <150 | 110–150 |
| AA (D.U.) | 75 a ± 1.06 | 69.6 b ± 0.67 | 56 c ± 0.36 | 79.2 a ± 0.57 | 75.6 b ± 0.58 | 78.5 ab ± 1.38 | 71.3 b ± 1.26 | 74.4 a ± 1.44 | 71.2 b ± 1.47 | 72 ± 0.46 | | 40–70 |
| FAN (mG/L) | 191 b ± 2.79 | 239 a ± 3.53 | 155 b ± 3.35 | 238 a ± 3.73 | 231 a ± 6.07 | 238 a ± 5.52 | 208 a ± 5.53 | 221 a ± 6.01 | 217 a ± 5.8 | 215 ± 1.88 | <150 | 140–190 |

[1] BA = Brewers Association Consensus Targets (2014). Provide guidelines for preferred malting barley characteristics for craft brewers. [2] AMBA = American Malting Barley Association Malting Barley Breeding Guidelines (AMBA, 2018). Provide ideal commercial malt criteria for "All Malt" Brewing. PLMP = kernel plumpness; FGDB = fine grind dry basis (malt extract); KI = soluble to total protein ratio (the Kolbach Index); FAN = free amino nitrogen; DP = diastatic power; AA = α-amylase.

The malting method influenced the Kolbach Index (KI), β-glucan, DP, alpha amylase and free amino nitrogen (FAN) (Table 2). Even with the effect of different malting regimes, genotype affected the most malt quality traits overall, including barley protein content, malt extract, KI, β-glucan, DP, alpha amylase and FAN (Table 2). While a smaller KI can indicate poorer modification, with less of the total protein converting to soluble protein in the wort, 117.17 was the only genotype to fall within the target range. All genotypes, including the control variety, exceeded the recommended amount of β-glucan, which is another indicator of poor modification (Table 1). Overall, genotype 117.17 appeared to satisfy more of the malt quality targets than the other genotypes. However, 117.17 had the highest β-glucan content, which can be more problematic for brewers compared to other malt quality traits because of dramatic effects on wort viscosity and subsequent brewing challenges.

**Table 2.** Three-way analysis of variance of malt method (modified or uniform), genotype (Copeland, 117.17, 120.14, 120.17), and replicate (1, 2, 3) and the interaction between malt method and genotype. *F*-values (*F*) and *p*-values (*p*) are reported for each malt quality traits listed. *p*-values less than alpha 0.05 and the respective *F*-value are bolded.

|  |  | Malt Method | Genotype | Replicate | Maltster * Genotype |
|---|---|---|---|---|---|
| Barley Protein (%, DB) | *F* | 1.9 | **11.3** | **22.9** | 0.032 |
|  | *p* | 0.19 | **<0.001** | **<0.0001** | 0.99 |
| Kernel Plumpness (on 6/64″) | *F* | 0.483 | 2.54 | 0.284 | 1.760 |
|  | *p* | 0.5 | 0.100 | 0.76 | 0.2 |
| Fine Grind Malt Extract (%) | *F* | 0.998 | **7.18** | **8.15** | 1.33 |
|  | *p* | 0.34 | **0.004** | **0.005** | 0.31 |
| Kolbach Index (%) | *F* | **24.1** | **71.4** | **14.2** | 2.24 |
|  | *p* | **<0.001** | **<0.0001** | **<0.001** | 0.13 |
| β-glucan (ppm) | *F* | **151** | **28.5** | 1.75 | **6.03** |
|  | *p* | **<0.0001** | **<0.0001** | 0.21 | **0.007** |
| Diastatic Power (°L.) | *F* | **56.4** | **15.7** | **16.6** | 0.642 |
|  | *p* | **<0.0001** | **<0.0001** | **<0.001** | 0.6 |
| Alpha Amylase (D.U.) | *F* | **52.4** | **217** | **8.21** | **21.2** |
|  | *p* | **<0.0001** | **<0.0001** | **0.004** | **<0.0001** |
| Free Amino Nitrogen (mG/L) | *F* | **129** | **91.7** | 3.68 | 2.2 |
|  | *p* | **<0.0001** | **<0.0001** | 0.052 | 0.13 |

* = interaction; bolded values denote a significant *p* value and the associated *F* value.

Considering the array of malt quality traits, a necessary balance must be achieved to ensure optimal malt quality. It could be argued that the small-scale, experimental malts would be unacceptable to craft brewers given the malt quality observed. However, in a white paper published by the BA in 2014, craft brewers ranked flavor above all other malt quality traits [21]. While guidelines exist for certain beer styles, and breeding for malt barley flavor has historically relied on a defect elimination process [51], there is a lack of consensus available for malt flavor guidelines. Moreover, fundamental tools have only recently been developed to aid in the evaluation of malt flavor, such as a base malt flavor map published by DraughtLab and the hot steep malt sensory method [25], which has been applied in several studies [29,30,52,53]. The question for craft maltsters and brewers may then become a matter of balancing desirable flavor characteristics with necessary malt quality parameters. It is possible that certain craft brewers may tolerate sub-optimal malt quality in favor of redeeming sensory qualities, arising from novel or distinctive flavor or other malt characteristics, as a means of product differentiation in an increasingly competitive market [15].

Grain quality, especially grain protein and moisture content, is a key indicator of potential malt quality [54]. While grain protein content ranged from 10.6% to 13.7% for the replicated field trial samples, genotype 117.17 and Copeland and field replicates one and three had a mean value within the target range (9.5–12.5%) (Table 1). Barley protein content is linked to extract and enzyme levels via the malting process and can have downstream effects on the brewing process. Higher protein content can lower the amount of malt extract available to the brewer, ultimately reducing alcohol production [55]. Conversely, insufficient protein content can impact FAN, which is an essential component of proper yeast nutrition, and enzyme content (e.g., DP and alpha amylase), which is required to hydrolyze starch to simple sugars [56,57]. Field replicate two had higher protein content, DP and alpha amylase, and lower KI than replicate one and three, and lower extract than replicate one (Table 1). There were no significant correlations between the malt quality traits and beer rank. However, an ordered probit model showed that moisture (coefficient = 4.50, *p* value $\leq$ 0.0001) and malt extract (coefficient = 0.84, *p* value $\leq$ 0.0001) had a positive effect on rank. This provides evidence for a potential cascade effect between grain protein content, extract, malt quality, brewhouse performance, and consumer acceptance.

Downstream effects of protein content on malt quality traits can be mediated. This can be accomplished by managing water absorption during the malting process with modified steep regimes. Alternatively, sampling fields and separating grain based on protein content before malting can mitigate difficulties in the malt house [58]. Low protein kernels uptake water more quickly compared to high protein kernels, leading to differences in germination [59]. Variable grain protein within a batch of grain can present a formidable challenge for maltsters, often leading to inconsistent malting and malted grain that can range from under-modified, to properly modified, to over-modified [27]. In this study, varying field conditions, such as differences in soil fertility (e.g., nitrogen availability), manifested as a significant effect of field replicate on protein content (Table 2). Moreover, we observed differential malt extract between the genotypes (e.g., 117.17 and Copeland) at similar grain protein content, which indicates a genotype interaction effect (Table 1). Consequently, malting at the commercial scale proved challenging (Table S2). This study illustrates the critical role of producer management and agronomics as a foundational component and fundamental driver of malt quality and brewing performance, through the production of high quality and uniform grain.

It is important to note that sensory evaluation of beers is best performed using properly modified malts that align with industry guidelines [27]. The commercial-scale malts used to produce the beers for the consumer sensory evaluation differed considerably with regard to their degree of modification, and can be characterized as under-modified (Table 3). Each commerically malted genotype had malt quality outside of industry guidelines, and genotype 117.17 seemed to be most impacted in terms of β-glucan content and enzyme levels (Figure 1, Table 3). Given the study design, evaluating the genotypes under commercial malting conditions and subsequently during the consumer sensory evaluation was essential to understanding their suitability to a regional grain system at scale and potential contributions to beer flavor.

**Table 3.** Malt quality data presented for the commercial-scale malts used to produce the beers for the consumer sensory evaluation.

| Malt Quality Trait | Genotype | | | | | |
|---|---|---|---|---|---|---|
| | Copeland | 120.14 | 120.17 | 117.17 | BA [1] | AMBA [2] |
| Moisture | 3.8 | 4.1 | 4.1 | 4.1 | | |
| Friability | 77.8 | 81.6 | 78.2 | 70.1 | | |
| FGDB (%) | 80.1 | 78.7 | 78.8 | 77.8 | >81.0% | |
| Color | 1.78 | 1.78 | 1.64 | 1.21 | | |
| β-glucan (ppm) | 382 | 229 | 364 | 735 | <140 | <100 |
| KI | 43.2 | 38.2 | 37 | 31.5 | 35–45 | 38–45 |
| FAN (mG/L) | 171 | 171 | 158 | 110 | <150 | 140–190 |
| DP (°L.) | 104 | 111 | 112 | 82 | <150 | 110–150 |
| AA (D.U.) | 43.4 | 52.5 | 46.6 | 24.1 | | 40–70 |

[1] BA = Brewers Association Consensus Targets (2014). Provide guidelines for preferred malting barley characteristics for craft brewers. [2] AMBA = American Malting Barley Association Malting Barley Breeding Guidelines (AMBA, 2018). Provide ideal commercial malt criteria for "All Malt" Brewing. FGDB = fine grind dry basis (malt extract); KI = soluble to total protein ratio (the Kolbach Index); FAN = free amino nitrogen; DP = diastatic power; AA = α-amylase.

Barley genotype contributions to beer flavor have been documented in a growing body of recent research [26,28–30,51,52]. Relating to malt modification, Herb et al. (2017) demonstrated there is a significant contribution of barley genotype to beer flavor even when over- and under-modified malts are intentionally produced and their impacts on beer flavor are considered. This supports the possibility that despite the varying degrees of modification, differentiation of the beers shown through the consumer sensory evaluation likely arose from the different barley genotypes used in this study. In their study, when levels of β-glucan decreased and malt extract increased, the trained panel detected an overall reduction in fruit and floral and an increase in malt, toasted, and related flavors. Moreover, Sayre-Chavez et al., 2022 [51] identified a genetic basis for barley contributions to beer

flavor, and state that while barley genotype contributions to beer flavor differ, differences in sensory attributes and metabolite profiles are not simply due to the degree of malt modification and/or difference in beer analytics. Additional studies should be conducted to enhance the understanding of how barley genotype, and barley genotype interactions with various factors related to the malting and brewing process, influence beer flavor.

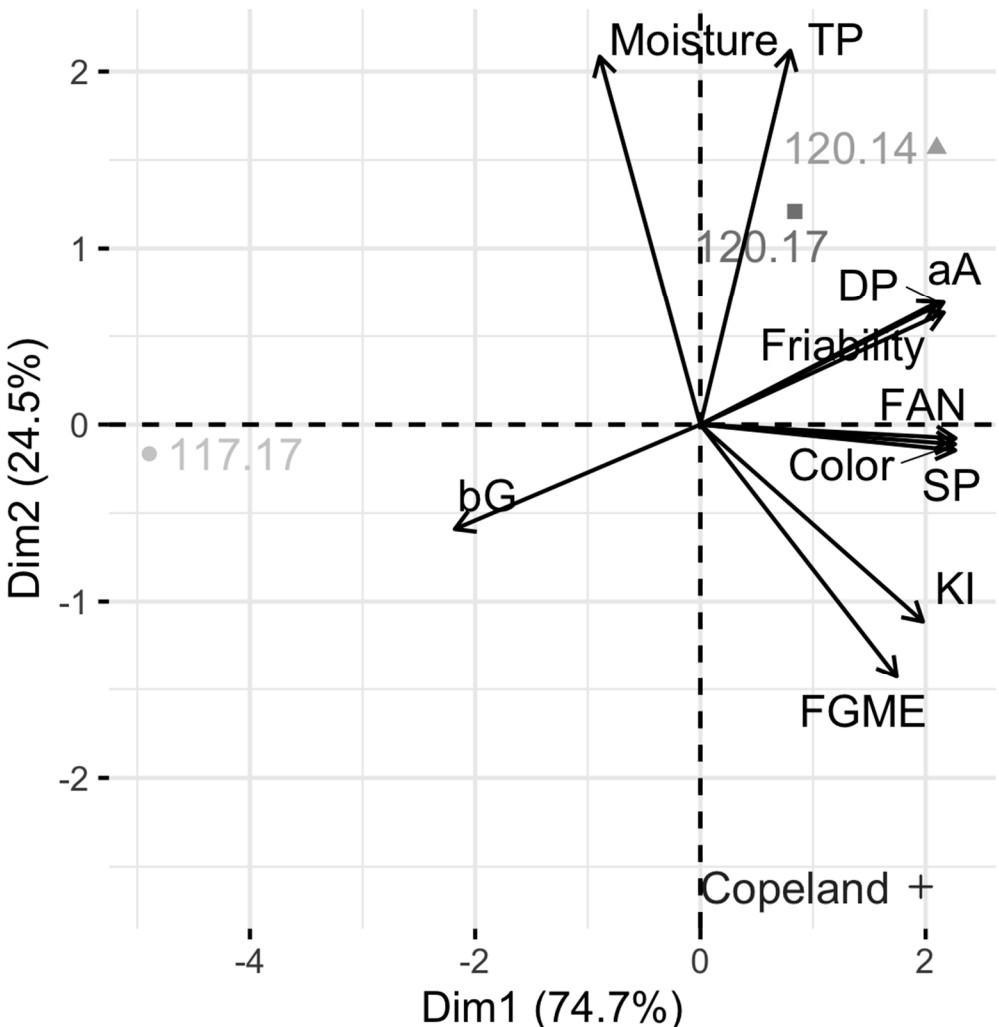

**Figure 1.** Principal component analysis of the commercial-scale malts used to produce the beers for the consumer sensory evaluation. Biplots shows the first principal component (Dim1) and second principal components (Dim2). Malt quality traits include moisture content (%), total protein (TP; %), diastatic power (DP; °L.), α-amylase (AA; D.U.), friability (%), free amino nitrogen (FAN; mG/L), wort color (color; °SRM), soluble protein (SP; %), the Kolbach index (soluble to total protein ratio; KI; %), fine grind malt extract (dry basis; FGME; %), and β-glucan content (ppm).

*3.3. Consumer Survey Responses Support Regional Grain System Development*

Consumers were asked several agreeing statements, and we are able to report a significant difference among responses for each question as well as the responses that are significantly different from the rest (Critical difference [Sheskin] procedure). Responses to "I have a strong preference for one style of beer (e.g., lager, IPA)" differed (Cochran's Q Test, Q Observed = 102.3, df = 8, $p < 0.0001$). Most consumers responded to this question with "Moderately agree" (30.4%) or "Agree very much" (27.5%). This provides important context for the results of this study, as this may have influenced consumer sensory perception and reporting behavior. For example, experience-based knowledge has been shown to be linked with familiar beers, compared to general knowledge being linked to unfamiliar beers [60].

Each consumer's sensory experience could have been impacted by the alignment of the test beers with their preferences or expectations.

The remaining agreeing statements focus on consumer purchasing behavior and regional grain system awareness. Responses to "It is important to me that beer is brewed with local ingredients" differed (Cochran's Q Test, Q Observed = 84.209, df = 8, $p < 0.0001$), and most consumers responded either that they "Neither Disagree or Agree" (26.1%) or "Moderately Agree" (21.0%). When asked if "I would pay more for a beer if I knew that it supported local farmers", responses also differed (Cochran's Q Test, Q Observed = 161.1, df = 8, $p < 0.0001$). Most of the consumers responded that they "Agree very much" (31.9%), "Moderately agree" (26.8%) or "Extremely agree" (24.6%). This indicates that local ingredients may not be as important to consumers, compared to supporting local farmers. This may seem paradoxical, as several studies have shown that consumers will pay more for local or organic products. However, these results strongly indicate where the value proposition may exist for consumers when interacting with a beer produced with local grains. By demonstrating how a more expensive beer can support local farmers, brewers may be able to create symbolic value for a beer that could translate to material value for farmers.

Finally, responses differed when consumers were asked "Because of this tasting project, I am more aware of the opportunity for local brewers to partner with local grain farmers" (Cochran's Q Test, Q = 140.7, df = 8, $p < 0.0001$). "Extremely Agree" had the highest number of consumer responses (31.9%). This demonstrates the potential of facilitated sensory experiences to support regional grain systems through outreach and education.

Translating locally grown craft malt into economic opportunity for farmers depends on numerous factors. These include available malting capacity, affordable grain transportation and malt distribution, a level of convenience for brewers commensurate to their current supply, awareness and willingness to pay (WTP) a price premium among brewers, and ability among brewers to pass along higher prices to consumers willing to pay more (whether for local ingredients, to support farmers, or other value proposition).

The high level of willingness (31.9% Agree very much) among consumers to pay more for beer that supports local farmers, coupled with the high level of enthusiasm we found among brewers for using locally grown malt, generally agrees with findings that craft malt does translate into better prices for farmers growing specialty malting barley, but within market limits and provided other logistical obstacles can be overcome. This supports the trend in the craft malting industry to pay farmers more ($0.37/kg) than the large-scale malt houses (0.18 USD/kg) [23,24].

### 3.4. Consumer Sensory Evaluation Reveals Barley Genotypes Influences on Beer Flavor

Consumers demonstrated the ability to differentiate beers made from genetically distinct barley genotypes through ranking, open description, and selection of attributes from a beer flavor map. Usually, sensory scientists choose hedonic scales over hedonic ranking. While hedonic scales can be more informative and provide parametric data that can be analyzed using a wider range of statistical techniques, a rank scale was selected for this study because it is an intuitive task and is perhaps easier for a consumer to complete compared to a hedonic scale, especially in the context of the other more difficult sensory tasks asked of the consumers in this study. Moreover, forced ranking has the potential to identify a most liked sample, especially if the samples are uniformly disliked or liked, which can be difficult to determine beforehand [61].

Genotype had a significant effect on beer liking (Friedman Test, Q = 8.809, df = 3, $p$ value = 0.032) (Figure 2). Post hoc analysis of rank data revealed that consumers liked genotype 120.17 (2.341 ± 1.097) more than genotype 117.17 (2.768 ± 1.135) (Nemenyi test, $p$ value = 0.032). Based on liking, either 120.14 (2.406 ± 1.098) or 120.17 could replace Copeland (2.486 ± 1.109) in beers. This corroborates the results of Craine et al., 2021 [30], where beers made from genotypes 120.14 and 120.17 had higher acceptance ratings than Copeland for appearance, taste/flavor, and overall liking. It may be possible for 117.17 to

replace Copeland in beers, but factors that may have influenced lower liking compared to 120.17 would have to be considered and controlled if possible.

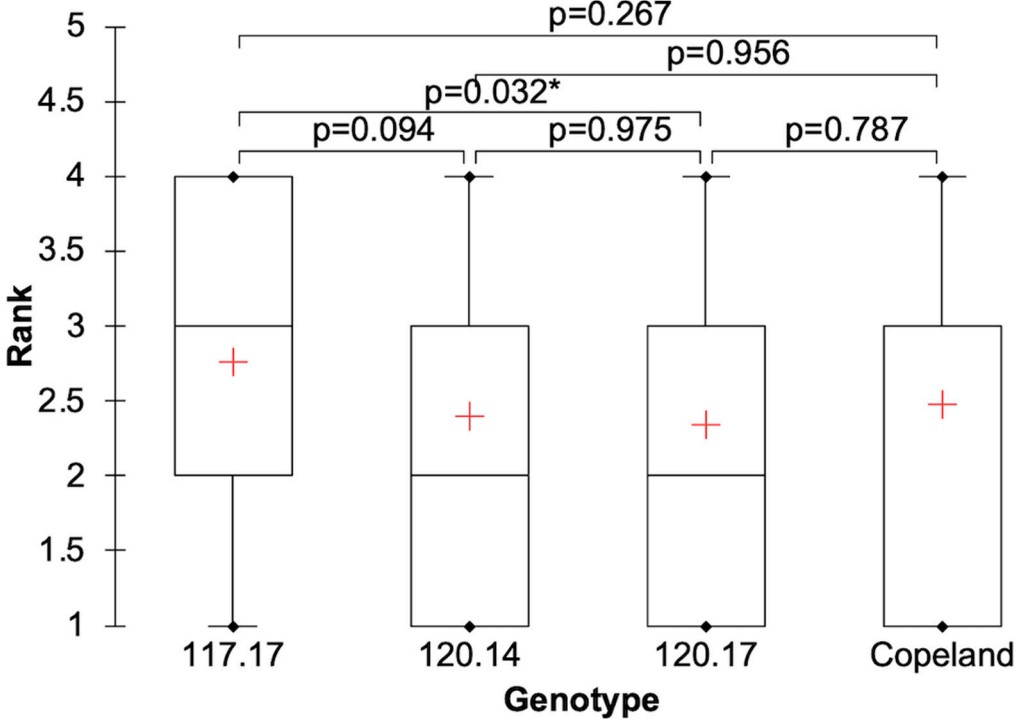

**Figure 2.** Boxplot representing rank data (1 = favorite, 4 = least favorite) for each genotype. One box plot per genotype is displayed. Red crosses correspond to mean values and central horizontal bars correspond to median values. Upper and lower limits of the box are the third and first quartiles, respectively. The box plot's horizontal width has no statistical meaning. *: significant at level alpha = 0.05; note: Copeland median value is 3.

To understand which attributes may have influenced beer liking, correlations between beer rank and attributes were analyzed for the four separate attribute data sets. These data sets included the open description coded according to the terms represented by the beer flavor map (Open Description), novel terms from the open description not included in the beer flavor map (Novel Terms), check all that apply using the beer flavor map (CATA), and CATA and open description terms combined (Combined) (Table 4). Several significant positive and negative correlations existed between the attributes and beer ranking. The highest ranked sample was given a 1, so a positive correlation is between the most highly ranked sample (i.e., a 1, not 4) and the attribute. Weak positive correlations existed between beer rank and *good flavor, floral, fruity, sweet aromatic, citrus* and *nutty*, while *sweet* and *smooth* had stronger positive associations (Table 4). Conversely, weak negative associations existed between rank and *sour, bitter, stale, bland, earthy,* and *drying*.

Sweetness and bitterness have been shown to increase and decrease liking, respectively, with light beers [62], among female consumers [63], and among Brazilian craft beers [46]. Craine et al., 2021 found that beer liking had the strongest negative associations with *chemical* and *stale*, and the strongest positive associations with *fruity, nutty, sweet aromatic, floral,* and *citrus*. Comparable results between these two studies, which evaluated nearly the same set of genotypes using different malting, brewing and sensory evaluation methods, provide evidence for how specific attributes may consistently influence consumer beer liking. However, it is important to note that these relationships are specific to the consumers and beers evaluated in these studies and may not extend to other populations and products.

**Table 4.** Significant correlations between beer rank based on overall liking and individual attributes in each of the four sensory attribute data sets. Spearman rank correlation coefficients are provided ($\rho$) along with *p* values.

| Data Set | Attribute | $\rho$ | *p* Value |
|---|---|---|---|
| CATA | fruity | 0.12 | 0.006 |
| | floral | 0.12 | 0.004 |
| | sweet | 0.25 | <0.0001 |
| | stale | −0.19 | <0.0001 |
| Novel Terms | good flavor | 0.13 | 0.003 |
| | smooth | 0.14 | <0.001 |
| | bland | −0.19 | <0.0001 |
| Open Description | smooth | 0.17 | <0.0001 |
| | sweet | 0.15 | 0.003 |
| | floral | 0.10 | 0.026 |
| | fruity | 0.12 | 0.006 |
| | citrus | 0.11 | 0.012 |
| | nutty | 0.09 | 0.037 |
| | earthy | −0.09 | 0.031 |
| | sour | −0.12 | 0.005 |
| Combined | stale | −0.19 | <0.001 |
| | drying | −0.08 | 0.049 |
| | sour | −0.12 | 0.004 |
| | bitter | −0.09 | 0.036 |
| | nutty | 0.09 | 0.044 |
| | floral | 0.12 | 0.007 |
| | fruity | 0.12 | 0.006 |
| | citrus | 0.10 | 0.019 |
| | sweet | 0.23 | <0.001 |
| | sweet aromatic | 0.12 | 0.005 |
| | smooth | 0.23 | <0.001 |

The data sets are "CATA" (check all that apply using the beer flavor map), "Novel Terms" (novel terms from the open description not included in the flavor map), "Open Description" (open description coded according to the flavor map) and "Combined" (CATA and open description combined).

The consumers further differentiated the beers using the open description and CATA tasks, as evidenced by significant differences among the genotypes within the attribute data sets (Table 5). While there was no significant association between genotypes and attributes for CATA ($\chi^2$ = 31.875, df = 48, *p* value = 0.965), Open Description ($\chi^2$ = 65.468, df = 96, *p* value = 0.993), Novel Terms ($\chi^2$ = 47.843, df = 45, *p* value = 0.358), and Combined ($\chi^2$ = 63.604, df = 66, *p* value = 0.561), Cochran's Q test for each attribute showed a significant difference among the genotypes for particular attributes within each data set, except for the Novel Terms data set (Table 5). For CATA, 120.14 had more *bitter* citations than 120.17. Moreover, 120.14 had more *smooth* citations than 117.17, and 117.17 had fewer *thin* citations than Copeland. For Open Description, 117.17 had more *sour* citations than 120.14 and more *woody* citations than 120.17. Finally, for the Combined data set, 117.17 had more *sour* and fewer *smooth* citations than 120.14, and more *dairy* citations than Copeland. In Craine et al. (2021), 117.17 was associated through correspondence analysis with *butter*, a secondary attribute within *dairy*. In another study of genotype-dependent beers evaluated by untrained panelists, Windes et al., 2021 found *citrus, floral, hoppy*, and *sweet* to be differentiating attributes. Their study employed CATA methodology, which other studies have shown to be an effective method for untrained panelists to differentiate beer samples [31,64–69]. This study aimed to expand on these studies by employing less restrictive, although perhaps more difficult sensory evaluation methodologies (i.e., open description and CATA). Even without prior training, the consumers in this study demonstrated an ability to differentiate the beer samples by performing challenging sensory evaluation tasks during a central location test. This study builds on an emerging body

of evidence that supports sensory evaluation of beer by untrained panelists, which relies on the increasingly discerning ability of consumers to communicate product experiences beyond basic preferences [70].

**Table 5.** Independent comparison of genotypes for each attribute using Cochran's Q test. Only attributes with significant *p* values (alpha = 0.05) are included. Multiple pairwise comparisons using the critical difference (Sheskin) procedure allows separation of the genotypes by attribute. Cell values represent the proportion of citation frequencies. Values within a row that do not share a letter are significantly different (bolded for emphasis).

| Data Set | Attribute | 117.17 | 120.14 | 120.17 | Copeland | *p* Value |
|---|---|---|---|---|---|---|
| CATA | bitter | 0.123 (ab) | **0.217 (b)** | **0.101 (a)** | 0.196 (ab) | 0.011 |
| | smooth | **0.080 (a)** | **0.188 (b)** | 0.130 (ab) | 0.145 (ab) | 0.017 |
| | thin | **0.036 (a)** | 0.051 (ab) | 0.065 (ab) | **0.101 (b)** | 0.028 |
| Open Description | sour | **0.138 (b)** | **0.029 (a)** | 0.065 (ab) | 0.065 (ab) | 0.004 |
| | woody | **0.080 (b)** | 0.036 (ab) | **0.007 (a)** | 0.051 (ab) | 0.027 |
| Combined | sour | **0.167 (b)** | **0.043 (a)** | 0.101 (ab) | 0.130 (ab) | 0.004 |
| | smooth | **0.123 (a)** | **0.246 (b)** | 0.210 (ab) | 0.181 (ab) | 0.015 |
| | dairy | **0.058 (b)** | 0.022 (ab) | 0.014 (ab) | **0.007 (a)** | 0.027 |

The data sets are "CATA" (check all that apply using the beer flavor map), "Open Description" (open description responses coded according to beer flavor map) and "Combined" (CATA and Description data sets aggregated and analyzed jointly). No significant differences were found for any of the attributes from the "Novel Terms" (novel terms from description provided by consumers) data set. Bolded values denote a significant *p* value and the associated *F* value.

## 4. Conclusions

This study augments an emerging body of evidence that highlights the important contributions of barley genotype to beer flavor, while demonstrating how differences between advanced barley breeding lines and a control variety can be traced from ground to glass. Varying responses of the barley genotypes to production conditions led to differing grain quality. Genotype effects extended to malt quality at the experimental and commercial scales, and ultimately influenced beer flavor. Compared to field replicate and malting method, genotype affected the most malt quality traits (seven out of eight). Responses to commercial-scale malting varied by genotype, resulting in malts that were generally under-modified, with breeding line 117.17 being impacted the most in terms of β-glucan content and enzyme content. Untrained panelists differentiated beers made from commercial malts of each barley genotype using multiple sensory evaluation tasks, which included ranking the beers based on overall liking, providing a description in their own words, and performing a check all that apply using a beer flavor map. The consumers ranked breeding line 117.7 lower than 120.17 based on overall acceptance, and successfully utilized the sensory tasks to substantiate differentiation. For example, 117.17 had fewer *smooth* and more *sour* citations than 120.14, more *dairy* citations than Copeland, and more *woody* citations than 120.17. Either 120.14 or 120.17 could replace the control (Copeland) in beers. Significant weak positive and negative correlations were found between rank and sensory attributes, which could inform breeding and brewing strategies. A considerable majority (91.3%) of consumers agreed that they would pay more for a beer if they knew that it would support local farmers, which recognizes the fundamental role barley growers play in supporting the craft malting and brewing industries. Further research should aim to build on this study, by investigating how barley genotype, and barley genotype interactions with various factors related to the production, malting and brewing, contribute to beer flavor.

**Supplementary Materials:** The following are available online at https://www.mdpi.com/article/10.3390/beverages8020030/s1, Table S1: Experimental malting conditions of field replicates by the uniform method (conducted by the United States Department of Agriculture [USDA] Cereal Crops Research Unit [CCRU], Madison, WI, USA) and the modified method (conducted by the Hartwick College Center for Craft Food and Beverage, Oneonta, NY, USA); Table S2: Commercial-scale malting conditions for the genotype, as performed by Gold Rush Malting, LLC. Baker City, OR, U.S.A; Table S3: Respondent characteristics from the consumer survey; Table S4. Respondent data from the sensory evaluation; Table S5. Malt quality data for each genotype, as malted by Gold Rush Malting LLC.; Table S6. Replicated (*n* = 3) grain and malt quality data for each genotype using the modified method; Table S7. Replicated (*n* = 3) grain and malt quality data for each genotype using the standard method.

**Author Contributions:** Conceptualization, K.M.M., S.B., E.B.C. and C.F.R.; methodology, K.M.M., S.B., E.B.C. and C.F.R.; formal analysis, E.B.C.; investigation, E.B.C. and S.B.; resources, K.M.M., S.B., E.B.C. and C.F.R.; data curation, S.B. and E.B.C.; writing—original draft preparation, S.B. and E.B.C.; writing— K.M.M., S.B., E.B.C. and C.F.R.; visualization, E.B.C.; supervision, K.M.M., S.B. and C.F.R.; project administration, K.M.M., S.B. and C.F.R.; funding acquisition, K.M.M. and S.B. All authors have read and agreed to the published version of the manuscript.

**Funding:** Funding was provided through the Washington State University Center for Sustaining Agriculture and Natural Resources (CSANR) Biologically Intensive Agriculture and Organic Farming (BIOAg) program (Project ID: 165). The Port of Olympia provided funding through their Economic Development Special Projects Program.

**Institutional Review Board Statement:** The study was conducted in accordance with the Declaration of Helsinki, and approved by the Institutional Review Board of Washington State University (IRB File Number 17775-001 approved 8 August 2019).

**Informed Consent Statement:** Not applicable.

**Data Availability Statement:** The data presented in this study are available in Supplementary Tables S4–S7.

**Acknowledgments:** Special thanks to Bill Reisinger and Brian Thompson for growing the barley for this project. We'd also like to thank the United State Department of Agriculture Cereal Crops Research Unit and Aaron Macleod of the Hartwick College Center for Craft Food and Beverage for micromalting and malt quality analysis, Gold Rush Malt LLC for craft malting, Top Rung Brewing Co. for craft brewing, and Beata Vixie and Ross' Sensory Science Center for assisting with the design of consumer panel. Aba Kiser, Kelvin Dam, Adam Peterson, Jennifer Post and Talia Feinberg helped conduct the consumer panel. The South Sound Community Farmland Trust and Thurston Conservation District contributed funding and support. This material is based upon work supported by the National Science Foundation Graduate Research Fellowship under Grant No. 1842493. As a disclaimer, any opinion, findings, and conclusions or recommendations expressed in this material are those of the authors and do not necessarily reflect the views of the National Science Foundation.

**Conflicts of Interest:** The authors declare no conflict of interest.

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
