# Peer review of "From Ground to Glass: Evaluation of Unique Barley Varieties for Craft Malting, Craft Brewing, and Consumer Sensory"

_beverages, doi:10.3390/beverages8020030_

Round 1

Reviewer 1 Report

The work done was well framed in the introduction and the methodologies used were well described. Results and discussion approach was pertinent and with a strong statistical support.

Comments to the Author

Lines 392-394: Why did authors not include sour and bitter in the weak negative associations?

Author Response

Lines 392-394: Why did authors not include sour and bitter in the weak negative associations?

Sour and bitter were omitted by accident and are now included in line 393.

Reviewer 2 Report

The authors do a very good job of not only describing the work but also putting the results into context. I think this will be of value to the brewing community.

However, I have two general comments that think will improve the manuscript. The first is that if each of the malts were found to be of poor quality how does or would this effect the consumers experience and do the authors feel that brewing with malt that were determined to be of sufficient quality according to the ASBC and BA standard effect the sensory outcome. And second, would it be more appropriate to being the discussion with malt and malt quality then transition to the consumer information? A rearrangement of the topic flow may also aid in addressing my previous question.

Author Response

The authors do a very good job of not only describing the work but also putting the results into context. I think this will be of value to the brewing community.

However, I have two general comments that think will improve the manuscript. The first is that if each of the malts were found to be of poor quality how does or would this effect the consumers experience and do the authors feel that brewing with malt that were determined to be of sufficient quality according to the ASBC and BA standard effect the sensory outcome. And second, would it be more appropriate to being the discussion with malt and malt quality then transition to the consumer information? A rearrangement of the topic flow may also aid in addressing my previous question.

Lines 474 address the decision of brewing with suboptimal malts, especially from the perspective of a brewer contributing to the definition of BA guidelines. Additionally, lines 522-531 justify the decision to use the malts. When preparing the manuscript, we found a lack of information in the literature that could provide insight into how changes in specific malt quality parameters could influence the consumer experience and consumer sensory outcome. Revisiting referenced papers did not reveal an example of changes in malt modification influencing sensory, as reported by a trained panel, and mention of these results is included in line 409-411. It is uncommon to use malt outside of quality standards, and this study provides a valuable example of potential relationships between suboptimal malt quality and consumer sensory. Had we used a measure of consumer acceptance or liking, we could have achieved a greater understanding. However, omitting this measure in the study design was intended to balance the request of information from consumers with their ability to provide information of sufficient quality for analysis.

To address the second comment, it certainly would be appropriate to rearrange the discussion. This is valuable feedback, and the structure of the discussion has been modified thanks to this suggestion. We hope that this change will provide useful information and context before discussion of the sensory evaluation results.

Reviewer 3 Report

The paper present a very interesting investigation.

Author Response

Many thanks for the feedback.